# Prostate Cancer Biochemical Recurrence Resulted Negative on [68Ga]Ga-PSMA-11 but Positive on [18F]Fluoromethylcholine PET/CT

Riccardo Laudicella [1,2,3,*], Flavia La Torre [1], Valerio Davì [1], Ludovica Crocè [1], Demetrio Aricò [4], Giuseppe Leonardi [1], Simona Russo [1], Fabio Minutoli [1], Irene A. Burger [2,3] and Sergio Baldari [1]

1 Nuclear Medicine Unit, Department of Biomedical and Dental Sciences and Morpho-Functional Imaging, University of Messina, 98125 Messina, Italy
2 Department of Nuclear Medicine, University Hospital Zürich, University of Zürich, 8091 Zürich, Switzerland
3 Department of Nuclear Medicine, Kantonsspital Baden, 5404 Baden, Switzerland
4 Department of Nuclear Medicine, Humanitas Oncological Centre of Catania, 95125 Catania, Italy
* Correspondence: riclaudi@hotmail.it

**Abstract:** For prostate cancer (PCa) biochemical recurrence (BCR), the primarily suggested imaging technique by the European Association of Urology (EAU) guidelines is prostate-specific membrane antigen (PSMA) positron emission tomography/computer tomography (PET/CT). Indeed, the increased detection rate of PSMA PET/CT for early BCR has led to a fast and wide acceptance of this novel technology. However, PCa is a very heterogeneous disease, not always easily assessable with the highly specific PSMA PET with around 10% of cases occuring without PSMA expression. In this paper, we present the case of a patient with PCa BCR that resulted negative on [68Ga]Ga-PSMA-11 PET/CT, but positive on [18F]Fluoromethylcholine (Choline) PET/CT.

**Keywords:** prostate cancer; PET; PSMA; choline; biochemical recurrence

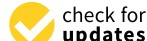



## 1. Introduction

Prostate cancer (PCa) is still the second most commonly diagnosed cancer in men [1]. Conventional imaging (ultrasound, magnetic resonance imaging—MRI) plays a fundamental role in PCa assessment, which could be magnified by positron emission tomography (PET) coupled with computed tomography (CT) or MRI.

Specifically, for PCa biochemical recurrence (BCR) the primarily suggested imaging technique by the European Association of Urology (EAU) guidelines is prostate-specific membrane antigen (PSMA) PET/CT, which has been demonstrated to be more sensitive compared to other radiopharmaceuticals [2,3].

Indeed, the increased detection rate of PSMA PET/CT for early BCR starting at prostate-specific antigen (PSA) levels of 0.2 ng/mL (while Choline PET/CT, able to assess the phospholidic metabolism [4], is recommended only at a PSA level of >1 ng/mL) has led to a fast and wide acceptance of this novel technology [5].

However, PCa is a very heterogeneous disease [6] and therefore not always easily assessable with the highly specific PSMA PET [7,8], with around 10% of cases occurring without PSMA expression.

In this paper, we present the case of a patient with PCa BCR that resulted negative on [68Ga]Ga-PSMA-11 PET/CT, but positive on [18F]Fluoromethylcholine (Choline) PET/CT.

## 2. Case

A 63-year-old patient was referred to our center for BCR of PCa. In 2015, he was diagnosed with clinically significant PCa (ISUP 3) and treated with radical prostatectomy (pT2cN1) and adjuvant pelvic radiotherapy (RT). Due to a fast PSA recurrence, in 2016

he underwent chemotherapy (Estramustine phosphate), followed by a period of stability. Between 2020 and 2021, a continuous increase in PSA values despite therapy was registered. At a PSA level of 3.05 ng/mL, he underwent a [68Ga]Ga-PSMA-11 PET that resulted negative (Figure 1a–c). However, at the co-registered low-dose CT there were 2 bilateral common iliac suspicious lymphnodes (max diameter 1.2 cm on the right side with no visible hilum) (**orange arrows**). Therefore, the patient was referred to [18F]Choline PET/CT 16 days later, which confirmed a high metabolic phospholipidic activity in the suspicious nodes (Figure 1d–f). According to the [18F]Choline PET/CT results the patient underwent an extended bilateral common iliac lymphadenectomy, with a following PSA drop (<0.01 ng/mL) in a personalized treatment approach. In Table 1, we also resumed the patient's PSA trend in correlation with main therapies.

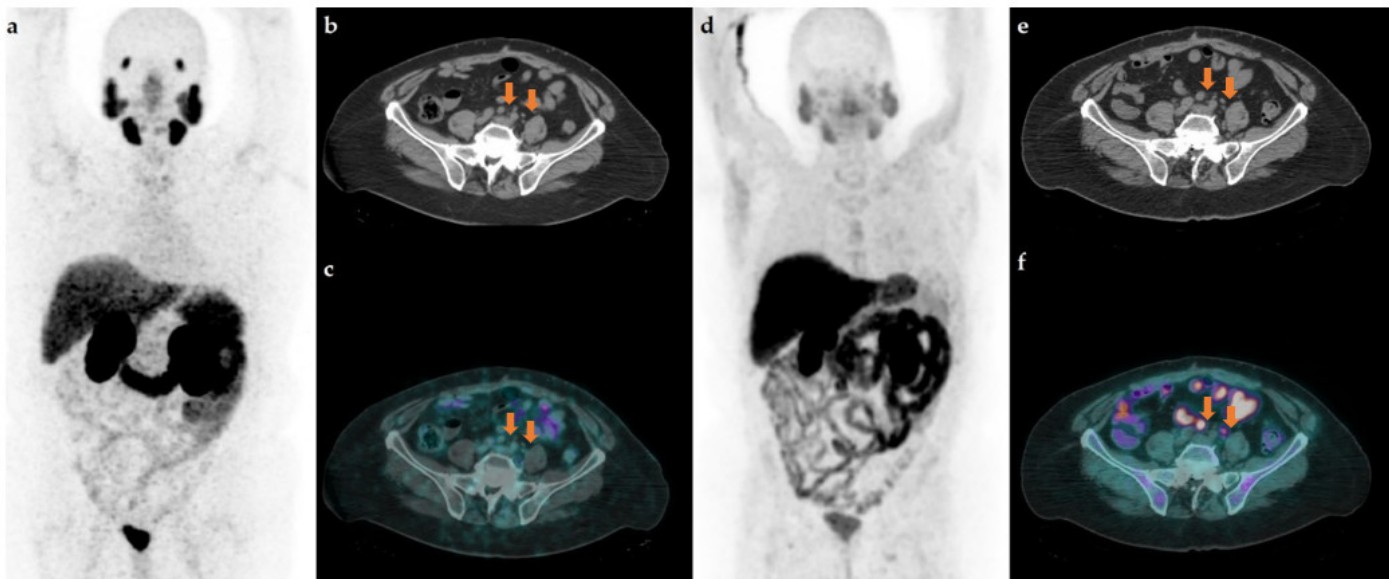

**Figure 1.** Maximum intensity projection (MIP) (**a**), axial low-dose CT (**b**) and fused [68Ga]PSMA-11 PET/CT (**c**); MIP (**d**), axial low-dose CT (**e**) and fused [18F]Choline PET/CT performed 16 days after PSMA PET/CT (**f**).

**Table 1.** PSA trend and main therapies.

| 01/2015 | 06/2015 | 12/2015 | 01/2016 | 01/2020 | 01/2021 | 03/2021 | 05/2021 | 06/2021 |
|---|---|---|---|---|---|---|---|---|
| RPE + pelvic RT | 0.25 ng/mL | 0.5 ng/mL | Estramustin ephosphate | 0.01 ng/mL | 1.9 ng/mL | 3.05 ng/mL | Extended bilateral common iliac limphadenectomy | <0.01 ng/mL |

*Legend*: PSA prostate-specific antigen; RPE radical prostatectomy; RT radiotherapy.

## 3. Discussion

In the molecular imaging scenario of PCa, several radiotracers are available: fluorodeoxyglucose (FDG) [9], fluciclovine [10], gastrin-releasing peptide receptor (GRPR) [11], Choline, PSMA, and also fibroblast-activating protein (FAP) [12].

However, currently, the most commonly available tracers in Europe are Choline and PSMA. PSMA is known to be expressed by most of the PCa lesions and therefore is more and more taking over the imaging indications of Choline PET in different settings [13–18].

In BCR, for PSA values below 0.5 ng/mL, [68Ga]Ga-PSMA PET/CT has a detection rate of 50% compared to 12.5% for [18F]Choline; for PSA values between 0.5–2.0 ng/mL, the detection rate is 70% and 30%, while for PSA values above 2.0 ng/mL the detection rate is 85% versus 60%, respectively [3].

Therefore, despite optimal results, the detection rate of PSMA PET/CT does not exceed 90% for PSA higher than 2 ng/mL, also encompassing the eventuality of reduced/absent PSMA expression in dedifferentiated PCa [19].

In this 10–15% "grey area", only one case report described and highlighted the added value of Choline PET to PSMA PET, particularly, in detecting seminal vesicle metastasis [20].

In our case, [18F]Choline PET/CT established the presence of high phospholipid activity in common iliac lymph nodes that were negative on [68Ga]Ga-PSMA-11 PET/CT.

Therefore, considering the heterogeneity of the disease and that almost 10% of PCa are PSMA-negative, in selected cases, we believe that choline PET/CT still represents an effective molecular imaging technique that should be considered by physicians.

## 4. Conclusions

Despite a well-known PSMA PET dominance in PCa assessment, Choline PET is still useful in selected cases (i.e., negative PSMA scans despite PSA > 1 ng/mL).

**Author Contributions:** Conceptualization, R.L. and S.B.; methodology, R.L. and I.A.B.; investigation, D.A. and R.L.; data curation, L.C. and G.L.; writing—original draft preparation, R.L., F.L.T. and V.D.; writing—review and editing, R.L. and I.A.B.; visualization, S.R.; supervision, I.A.B., F.M. and S.B.; project administration, R.L. All authors have read and agreed to the published version of the manuscript.

**Funding:** This research received no external funding.

**Institutional Review Board Statement:** All procedures performed involving the human participant were under the ethical standards of the institutional and/or national research committee and with the 1964 Helsinki declaration and its later amendments or comparable ethical standards.

**Informed Consent Statement:** Informed consent was obtained from the patient.

**Data Availability Statement:** Data are available for bona fide researchers who request it from the authors.

**Conflicts of Interest:** The authors declare no conflict of interest.

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
