# Peer review of "Prostate Cancer Biochemical Recurrence Resulted Negative on [68Ga]Ga-PSMA-11 but Positive on [18F]Fluoromethylcholine PET/CT"

_tomography, doi:10.3390/tomography8050205_

Round 1

Reviewer 1 Report

The manuscript presents an interesting case report showing the better prognostic efficacy of choline PET compared to the most acceptable PSMA PET. Though, the introduction of the manuscript is very short and should be elaborated more, especially on the use and mechanism of detection of [18F]Fluoromethylcholine 23 (Choline) PET/CT. The Case section also needs to be explained in more detail.  

Author Response

We sincerely thank both reviewers for their dedication to our manuscript. Here’s our point-by-point responses. Any modifications in the manuscript will be highlighted in yellow. 

Reviewer 1 

R1: The introduction of the manuscript is very short and should be elaborated more, especially on the use and mechanism of detection of [18F]Fluoromethylcholine (Choline) PET/CT. The Case section also needs to be explained in more detail.  

A1: Thank you for these suggestions. We improved the manuscript accordingly briefly explaining that Choline PET is able to assess the phospholipidic metabolism (ref. 4); we also improved the case by adding table 1 resuming the patient’s PSA trend also in correlation with main therapies.

Reviewer 2 Report

The authors describe clearly a case report of a male patient aged 63 years with recurrent prostate cancer. They give all needed information for diagnosis, treatment, and outcome. The  discussion provide context and any necessary explanation of specific decision for an additional PET/CT scan with the radiotracer [18F]Fluoromethylcholine 16 days after administartion of classical [68Ga]Ga-PSMA-11. The conclusion briefly outline the take-home message that depending on the blood PSA levels that radiotracer [68Ga]Ga-PSMA-11 alone can give false negative results which could lead to less effective treatment of the patients and that other tracer should be considered further.

 Minor things:

-         Fairly all abbreviations (CT, PET, etc.) have been explained/written out at the beginning of the article. Please provide this explanation also for EAU (line 18 & 33) and PSA (line 36).

-         Is it possible to give a reference for the statement starting in line 36: „the increased detection rate of PSMA PET/CT for early BCR starting at PSA levels 36 of 0.2 ng/ml (while Choline PET/CT is recommended only at a PSA level of > 1 ng/ml) has 37 led to a fast and wide acceptance of this novel technology.“

-         Personally would whish for a table showing the PSA levels of the individual patient starting from diagnosis in 2015 to final treatment.

Author Response

We sincerely thank both reviewers for their dedication to our manuscript. Here’s our point-by-point responses. Any modifications in the manuscript will be highlighted in yellow. 

Reviewer 2 

R2.1: Fairly all abbreviations (CT, PET, etc.) have been explained/written out at the beginning of the article. Please provide this explanation also for EAU (line 18 & 33) and PSA (line 36). 

R2.2: Is it possible to give a reference for the statement starting in line 36: „the increased detection rate of PSMA PET/CT for early BCR starting at PSA levels of 0.2 ng/ml (while Choline PET/CT is recommended only at a PSA level of > 1 ng/ml) has led to a fast and wide acceptance of this novel technology.“ 

R2.3: Personally would whish for a table showing the PSA levels of the individual patient starting from diagnosis in 2015 to final treatment. 

A2.1-2-3: Thank you for these suggestions. We added EAU and PSA acronyms’ spelling, added a reference as requested (ref. 5) and table 1 resuming the patient’s PSA trend also in correlation with main therapies.